# SOLVING INVERSE PROBLEM WITH UNSPECIFIED FORWARD OPERATOR USING DIFFUSION MODELS

## ABSTRACT

Diffusion models have excelled in addressing a variety of inverse problems. Nevertheless, their application is restricted by the requirement for specific prior knowledge of the forward operator. This paper presents a novel approach, *UFODM*, which circumvents this constraint by selecting the appropriate forward measurement, making the method more applicable to real-world scenarios. Specifically, our approach enables the concurrent estimation of both the reconstructed image and the characteristics of the forward operator during the inference stage. Our method effectively tackles inverse problems such as blind deconvolution, JPEG restoration, and super-resolution. Furthermore, we demonstrate the versatility of our approach in solving generic inverse problems through the automated selection of forward operators. Empirical evidence suggests that our framework has the potential to enhance the efficacy of diffusion models and extend their applicability in solving real-world inverse problems.

## 1 INTRODUCTION

Inverse problems are pervasive in fields like image processing and computer vision, where the objective is to recover an original image from distorted observations modulated by a forward operator. These problems manifest across a broad spectrum of applications, from generic image processing tasks such as deblurring, denoising, inpainting, and super-resolution, to specialized use-cases in medical imaging technologies like computed tomography and magnetic resonance imaging.

Diffusion models have recently gained prominence as an alternative to conventional optimization algorithms and Generative Adversarial Networks (GANs) for solving inverse problems. These models leverage stochastic processes to iteratively refine image reconstruction, obviating the need for an explicitly defined inverse operator — a requisite in deterministic methods. This inherent flexibility allows diffusion models to adapt to a variety of distortions and noise, making them an indispensable tool for a wide range of image processing tasks (Kawar et al., 2022; Chung et al., 2023b; Song et al., 2023; Wang et al., 2022; Chung et al., 2023a; Murata et al., 2023).

However, diffusion models have limitations. One significant shortcoming is their dependency on a well-characterized forward operator and its associated parameters (Kawar et al., 2022; Chung et al., 2023b; Song et al., 2023; Wang et al., 2022). In cases where the forward operator is not explicitly defined, the models necessitate at least a predefined functional form (Chung et al., 2023a; Murata et al., 2023). This constraint curtails their utility in environments where the properties of the forward operator are either ambiguous or not defined. Specifically, this limitation becomes a bottleneck in real-world applications with complex, unstructured settings, where the lack of clear specifications for the forward operator impairs effective signal reconstruction.

In this paper, we introduce a novel diffusion-centric framework designed for solving inverse problems, effectively circumventing the conventional constraints inherent to this field. Our methodology autonomously identifies the most fitting forward measurement from a pool of candidates, thereby introducing and formalizing the concept of "inverse problems with unspecified forward operator" within a Bayesian framework. In this setup, all relevant entities—namely the original image, the forward operator, and associated parameters—are conceptualized as latent variables, observable only through their corresponding measurements. Figure 1 offers a clear contrast, highlighting how our approach deviates fundamentally from existing diffusion-based solvers in addressing complex inverse problems without the pre-defined forward operator.

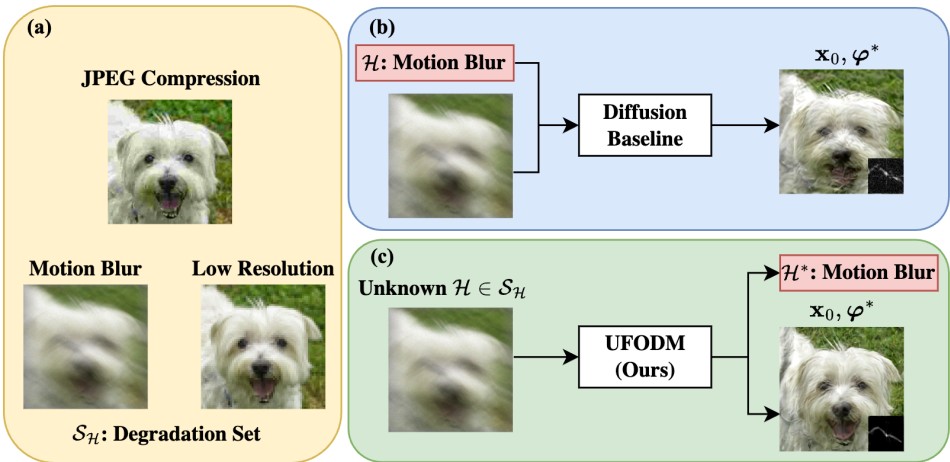

Figure 1: The comparison between our UFODM and other diffusion baselines (i.e. BlindDPS, GibbsDDRM). (a) The degradation set, including motion blur, JPEG compression and low resolution, (b) other diffusion baseline solving the blind inverse problem, which requires the specific forward operator information, (c) Our UFODM, simultaneous estimation of both the restored image and the forward operator's parameters, then automatically select the proper forward operator form.

Building upon the cornerstone methodology of partially collapsed Gibbs sampling (Van Dyk & Park, 2008), we introduce an innovative approximation algorithm that progressively enhances the accuracy of latent variable estimates. The anchor image is generated using a pretrained diffusion model, steered by these latent variables. To bolster computational performance, Maximum a Posteriori (MAP) estimation is harnessed to fine-tune both the forward operator and its associated parameters. Though devoid of formal asymptotic convergence proofs, our approach, dubbed UFODM, demonstrates compelling empirical support for its efficacy and robustness in tackling intricate inverse problems characterized by an unspecified forward operator.

To corroborate the validity of our approach, we conduct comprehensive evaluations using the well-established FFHQ (Karras et al., 2019), AFHQ (Choi et al., 2020) and ImageNet (Russakovsky et al., 2015) datasets. Initially, we position UFODM in direct competition with current state-of-the-art benchmarks (Murata et al., 2023), focusing on the challenging task of blind deconvolution. According to universally accepted evaluation metrics, such as FID, LPIPS, and PSNR, UFODM not only maintains parity but often surpasses existing solutions. We extend our inquiry into the complex domain of inverse problems with unspecified forward operator, experimenting with more prevalent forward mechanisms like JPEG compression, pixel-level mask out, and reduced resolution. Remarkably, our empirical analyses show that UFODM consistently identifies these elusive forward operators across diverse datasets, even outperforming cases with known operators. These persuasive results underline UFODM's potential as an innovative tool for surmounting current limitations in diffusion models, thus paving the way for future breakthroughs in image restoration and enabling its application to a broader array of real-world scenarios.

The main contributions of our work are summarized as follows:

- We conceptualize and mathematically formulate the novel notion of inverse problems with unspecified forward operators within a Bayesian framework. This involves devising a novel approximate inference technique, leveraging partially collapsed Gibbs sampling and maximum a posteriori inference, to alternately update latent variables.

- We architect a new computational paradigm, denoted as UFODM, capable of autonomously identifying the optimal forward measurement. This mitigates the inherent constraints tied to reliance on predefined forward operators in existing diffusion models.

- Our rigorous empirical assessment, utilizing FFHQ, AFHQ, and ImageNet datasets, substantiates that UFODM is proficient at inferring unspecified forward operators. The method attains a performance parity with known operators, thereby considerably broadening the diffusion models' realm of applicability in practical settings.

## 2 RELATED WORK

**Task-specific Methods**  The diffusion model has showcased outstanding capabilities, notably in resolving inverse problems such as super-resolution, inpainting, and deblurring, among others. Standard approaches (Saharia et al., 2022b;a; Whang et al., 2022) entail training a task-specific conditional diffusion model to grasp the mapping between the restored and the measurement images. However, this supervised method exhibits certain drawbacks, as it requires training distinct models for different tasks, potentially limiting their generalization beyond the specific settings where they were trained. Consequently, the models' generalization capacity may prove inadequate, possibly leading to failures in unfamiliar circumstances.

**Zero-Shot Methods**  An alternative widely used strategy in resolving inverse problems involves employing diffusion models as priors. These methods typically employ pre-trained diffusion models, eliminating the need for extra training. For instance, Denoising Diffusion Restoration Models (DDRM) (Kawar et al., 2022) utilizes singular value decomposition (SVD) to decompose the degradation operators, and then progressively denoises a sample to achieve the desired output. DPS (Chung et al., 2023b) broadens the application of diffusion solvers to efficiently manage general noisy (non)linear inverse problems by approximating the posterior sampling process. Unlike many existing approaches that are limited to linear inverse problems, Pseudoinverse-guided Diffusion Models (ΠGDM) (Song et al., 2023) can handle inverse problems with noisy, non-linear, or even non-differentiable measurements. ΠGDM leverages problem-agnostic models to estimate conditional scores from the measurement model of an inverse problem, and it has demonstrated competitive performance with state-of-the-art diffusion models trained on specific tasks.

Despite the progress made by the prior work in broadening the range of solvable inverse problems, they continue to heavily rely on substantial prior knowledge of the specific forward operator and known parameters, which limits their applicability in downstream tasks.

**Blind Inverse Problems**  Recent developments have advanced the exploration of inverse problems further, focusing on a blind setting where the forward operator's parameter is unknown, a situation termed the *blind inverse problem*. BlindDPS (Chung et al., 2023a) formulates an additional diffusion prior for the forward operator. The concurrent estimation of the forward operator parameters and the image is achieved via a parallel reverse diffusion process. However, a significant limitation of BlindDPS is the necessity to train another diffusion model for the parameter prior, leading to inefficiency and significantly curtailing its practical applicability. Conversely, GibbsDDRM (Murata et al., 2023) adopts a different strategy, not depending on a data-driven prior model of the measurement process. As an extension of DDRM (Kawar et al., 2022), GibbsDDRM constructs a joint distribution that includes the data, measurements, and linear operators. Using a pre-trained diffusion model as the data prior, the problem is resolved through posterior sampling, employing an efficient variant of a Gibbs sampler. Recently, Fast Diffusion EM (Laroche et al., 2023) employs the Expectation-Maximization (EM) algorithm to approximate the log-likelihood of the inverse problem using samples drawn from a diffusion model and a maximization step to estimate unknown model parameters. However, an additional trained denoiser is needed as a Plug & Play regularization. This task-specific design, while effective for its intended purpose, may limit its applicability to other inverse problems.

While these methods (Chung et al., 2023a; Murata et al., 2023; Laroche et al., 2023) have shown promising results in managing blind inverse problems, it is crucial to recognize that the blind setting remains distant from real-world scenarios. When the function form of the forward operator is unspecified, these methods lack guidance on how to handle the selection of the function form, presenting a challenge in practical applications.

## 3 METHOD

### 3.1 BACKGROUND

**Inverse Problem**  An inverse problem can be generally defined as the process of recovering the unknown image $\mathbf{x}$ from a known measurement $\mathbf{y}$ and a forward operator $\mathcal{H}_\varphi$ parameterized by $\varphi$. Mathematically, this process can be represented as

$$\mathbf{y} = \mathcal{H}_\varphi \mathbf{x} + \mathbf{z} \tag{1}$$

where $\mathbf{y}$ denotes the measurement image, $\mathbf{x}$ represents the clean image data that needs to be estimated, and $\mathbf{z} \sim \mathcal{N}\left(\mathbf{0}, \sigma_{\mathbf{y}}^2 \mathbf{I}\right)$ is an *i.i.d* additive Gaussian noise with known variance $\sigma_{\mathbf{y}}^2$. While solving inverse problems, the primary objective is to recover the original clean image from the measurement $\mathbf{y}$ and the specified forward function with known parameters. However, the effectiveness of these methods is often limited by the strong prior knowledge required. To address this, researchers have explored blind inverse problems (Chung et al., 2023a; Murata et al., 2023), where the parameter $\varphi$ of the forward operator $\mathcal{H}$ is unknown. In such cases, the parameter and the restored image must be estimated simultaneously.

**Diffusion Models**  Diffusion models are a class of latent variable generative models that have recently demonstrated state-of-the-art performance for image synthesis. Given a clean image sampled from the real image distribution $\mathbf{x}_0 \sim q(\mathbf{x}_0)$, the forward process of the diffusion model defines a fixed Markov chain:

$$q(\mathbf{x}_{1:T} \mid \mathbf{x}_0) = \prod_{t=1}^{T} q(\mathbf{x}_t \mid \mathbf{x}_{t-1}), \qquad q\left(\mathbf{x}_t \mid \mathbf{x}_{t-1}\right) = N\left(\mathbf{x}_t; \sqrt{1 - \beta_t}\mathbf{x}_{t-1}, \beta_t \mathbf{I}\right), \qquad (2)$$

The reverse process of the diffusion model iteratively removes the noise added in the forward process to generate a clean image in $T$ timesteps, formulated as a Markov chain with parameterized Gaussian transitions.

$$p\left(\mathbf{x}_{0:T}\right) = p(\mathbf{x}_T) \prod_{t=1}^{T} p(\mathbf{x}_{t-1} \mid \mathbf{x}_t), \qquad p_\theta\left(\mathbf{x}_{t-1} \mid \mathbf{x}_t\right) = N\left(\mathbf{x}_{t-1}; \mu_\theta\left(\mathbf{x}_t, t\right), \sigma_t^2\left(\mathbf{x}_t, t\right) \mathbf{I}\right) \quad (3)$$

Diffusion models are trained by optimizing the variational bound of negative log-likelihood $\mathbb{E}[-\log p_\theta(\mathbf{x}_0)]$, which enables them to capture the data distribution and learn a generative prior of the training data. In denoising diffusion probabilistic models (DDPM) (Ho et al., 2020), $\sigma_t\left(\mathbf{x}_t, t\right)$ are set as time-dependent constants $\sigma_t \mathbf{I}$. The parameterization of $\mu_\theta$ is a linear combination of $\mathbf{x}_t$ and $\epsilon_\theta(\mathbf{x}_t, t)$, where $\epsilon_\theta(\mathbf{x}_t, t)$ is a function that predicts the noise component of a noised sample $\mathbf{x}_t$. The parameters of $\mu_\theta\left(\mathbf{x}_t, t\right)$ are learned by optimizing the variational bound of negative log-likelihood $\mathbb{E}[-\log p_\theta(\mathbf{x}_0)]$. To achieve this, the training objective $\mathcal{L}_{\text{simple}}$ is simplified to a mean-squared error loss between the actual noise $\epsilon \sim \mathcal{N}(0, \mathbf{I})$ in $\mathbf{x}_t$ and the predicted noise.

$$\mathcal{L}_{\text{simple}} = ||\epsilon_\theta(\mathbf{x}_t, t) - \epsilon||^2 \qquad (4)$$

## 3.2 Inverse Problem with Unspecified Function: a Bayesian Perspective

In this paper, we focus on a more realistic setting termed as *inverse problems with unspecified forward operator*. In particular, given the measurement image $\mathbf{y}$ and a set $\mathcal{S}_{\mathcal{H}} = \{\mathcal{H}^1, \mathcal{H}^2, ..., \mathcal{H}^n\}$ of all potential forward operators, our goal is to estimate the clean image and the exact forward operator along with the corresponding estimated parameters. Naturally, this setup is more challenging than the traditional (blind) inverse problem, as it involves an unspecified forward operator and a significantly larger solution space.

If no additional assumptions are made, the defined problem is ill-posed. Specifically, we assume that $x_0$ is generated from an unknown distribution, which can be well approximated by a model distribution $p_\theta(x_0)$, and that the parameter $\varphi$ is drawn independently from a known prior $p(\varphi)$ that is not dependent on the data. The forward mapping function is chosen from a set of candidate forward operators. Formally, we define a joint distribution that encompasses the restored image $\mathbf{x}_0$, the measurement image $\mathbf{y}$, the forward operator $\mathcal{H}$, and their corresponding parameters $\varphi$, as follows:

$$p\left(\mathbf{x}_0, \mathcal{H}, \varphi, \mathbf{y}\right) = p\left(\mathbf{y} \mid \mathbf{x}_0, \mathcal{H}, \varphi\right) p\left(\varphi \mid \mathcal{H}\right) p\left(\mathbf{x}_0\right) p\left(\mathcal{H}\right). \qquad (5)$$

Here, $p\left(\mathbf{y} \mid \mathbf{x}_0, \mathcal{H}, \varphi\right)$ follows a Gaussian distribution $\mathcal{N}\left(\mathbf{y} \mid \mathcal{H}_\varphi \mathbf{x}_0, \sigma_{\mathbf{y}}^2 \mathbf{I}\right)$ according to the definition of the inverse problem (Eq. 1) and $p(\mathbf{x}_0)$ represents the prior probability of the data. In this paper, we leverage a pre-trained diffusion model, where the data prior is defined as $p_\theta(\mathbf{x}_{0:T}) = p_\theta(\mathbf{x}_T) \prod_{t=1}^{T} p_\theta(\mathbf{x}_{t-1} \mid \mathbf{x}_t)$. It is worth noting that the conditional distribution of the parameters, $p\left(\varphi \mid \mathcal{H}\right)$, is a simple distribution (e.g., uniform) and depends on the specific operator (see details in Sec. 4.1). Furthermore, we model the prior distribution over possible operators as a simple uniform distribution, denoted by $p\left(\mathcal{H}\right)$.

As detailed in the following sections, we perform inference for the posterior distribution $p\left(\mathbf{x}_0, \mathcal{H}, \boldsymbol{\varphi} \mid \boldsymbol{y}\right)$, which encapsulates the uncertainty in the estimation of the clean image, the forward operator, and the corresponding parameters, to solve *the inverse problem with unspecified forward operator* from a Bayesian perspective.

### 3.3 INFERENCE FOR THE TARGET POSTERIOR DISTRIBUTION

Taking inspiration from GibbsDDRM (Murata et al., 2023), we facilitate the alternative inference of the latent variables $\mathbf{x}_{1:T}$, the forward operator $\mathcal{H}$, and the corresponding parameters $\boldsymbol{\varphi}$, conditioned on all the other variables, from the posterior distribution. However, for the sake of simplicity and efficiency, we do not perform a Gibbs sampling as in prior work (Murata et al., 2023). Instead, we sample $\mathbf{x}_t$ by utilizing the diffusion model similarly but obtaining a Maximum a posteriori (MAP) estimate for both the function form $\mathcal{H}$ and its parameters $\boldsymbol{\varphi}$. As such, our designed algorithm is not guaranteed to converge in the asymptotic case theoretically while it indeed serves as an efficient and effective approximate inference solution for this challenging problem in practical scenarios (as supported by evidence in Sec. 4). We now present the alternative inference procedure in detail.

**Inference of $\mathbf{x}_t$.** We sample from $p_\theta(\mathbf{x}_0 \mid \mathbf{y}, \boldsymbol{\varphi}, \mathcal{H})$ via a guided diffusion process (Song et al., 2023) by defining $p_\theta(\mathbf{x}_{t-1} \mid \mathbf{x}_t, \mathbf{y}, \boldsymbol{\varphi}, \mathcal{H})$ for $t = T', ..., 1$ where $T' < T$ is a hyperparameter and $\mathbf{x}_{T'}$ is initialized by perturbing the measurement $\mathbf{y}$ with Gaussian noise at timestep $T'$. Rather than starting with random noise, the perturbed image of measurement $\mathbf{y}$ provides valuable semantic information as a reliable initialization. Given the forward operator $\mathcal{H}$ and the parameters $\boldsymbol{\varphi}$, the corresponding score can be decomposed via Bayes' rule:

$$\nabla_{\mathbf{x}_t} \log p_t\left(\mathbf{x}_t \mid \mathcal{H}, \boldsymbol{\varphi}, \mathbf{y}\right) = \nabla_{\mathbf{x}_t} \log p_t\left(\mathbf{x}_t \mid \mathcal{H}, \boldsymbol{\varphi}\right) + \nabla_{\mathbf{x}_t} \log p_t\left(\mathbf{y} \mid \mathbf{x}_t, \mathcal{H}, \boldsymbol{\varphi}\right), \quad (6)$$

where the first term can be approximated with the noise prediction network in the diffusion model, and the second term serves as a guidance component. According to Eq. 3 and Eq. 6, the $\mathbf{x}_{t-1}$ is sampling from the following Gaussian distribution given $\mathbf{x}_t$:

$$p_\theta(\mathbf{x}_{t-1} \mid \mathbf{x}_t, \mathbf{y}, \boldsymbol{\varphi}, \mathcal{H}) = \mathcal{N}\left(\mu_\theta\left(\mathbf{x}_t\right) + r_t^2 \Sigma_\theta\left(\mathbf{x}_t\right) \nabla_{\mathbf{x}_t} \log p_t\left(\mathbf{y} \mid \mathbf{x}_t, \boldsymbol{\varphi}, \mathcal{H}\right), \Sigma_\theta\left(\mathbf{x}_t\right)\right). \quad (7)$$

To handle nonlinear and non-differentiable operators (e.g., JPEG), we follow $\Pi$GDM (Song et al., 2023) to approximate the guidance term as:

$$\nabla_{\mathbf{x}_t} \log p_t\left(\mathbf{y} \mid \mathbf{x}_t, \boldsymbol{\varphi}, \mathcal{H}\right) \approx \left(\left(\mathbf{y} - \mathcal{H}\hat{\mathbf{x}}_t\right)^\top \left(r_t^2 \mathcal{H}\mathcal{H}^\top + \sigma_\mathbf{y}^2 \boldsymbol{I}\right)^{-1} \mathcal{H}\frac{\partial \hat{\mathbf{x}}_t}{\partial \mathbf{x}_t}\right)^\top. \quad (8)$$

where $\hat{\mathbf{x}}_t$ is the estimation of $\mathbf{x}_0$ given $\mathbf{x}_t$, and this guidance term is a vector-Jacobian product and can be computed with backpropagation.

**Inference of $\boldsymbol{\varphi}$.** We can sample from $\boldsymbol{\varphi}$ from the conditional distribution $p_\theta(\boldsymbol{\varphi} \mid \mathbf{y}, \hat{\mathbf{x}}_t, \mathcal{H})$. However, for efficiency and simplicity, we instead obtain the MAP estimation as follows:

$$\boldsymbol{\varphi}^* = \arg\max_{\boldsymbol{\varphi}} p_\theta(\boldsymbol{\varphi} \mid \mathbf{y}, \hat{\mathbf{x}}_t, \mathcal{H}) = \arg\min_{\boldsymbol{\varphi}} \|\mathcal{H}_{\boldsymbol{\varphi}}(\hat{\mathbf{x}}_t) - \mathbf{y}\|_2 + R_{\boldsymbol{\varphi}}(\boldsymbol{\varphi}), \quad (9)$$

where the first term and second term in the right optimization problem correspond to the likelihood of $\mathbf{y}$ and prior of $\varphi$ in the joint distribution (see Eq. 5) respectively. For differentiable operators like motion blur, we solve the optimization problem with gradient descent. Otherwise, we can perform zero-order optimization (Chen et al., 2017) for $\varphi^*$. See Sec. 4.1 for optimization details.

**Inference of $\mathcal{H}$.** Similarly to the inference of $\boldsymbol{\varphi}$, we perform MAP inference for $\mathcal{H}$ as follows:

$$\mathcal{H}^* = \arg\max_{\mathcal{H}} p(\mathcal{H} \mid \mathbf{x}_0, \mathbf{y}, \boldsymbol{\varphi}) = \arg\max_{\mathcal{H}} p(\mathbf{y} \mid \mathbf{x}_0, \boldsymbol{\varphi}, \mathcal{H}) p(\boldsymbol{\varphi} \mid \mathcal{H}) p(\mathbf{x}_0), \quad (10)$$

where the prior of $\mathcal{H}$ is omitted since it is a constant. For simplicity and efficiency, we do not infer $\mathcal{H}$ during the iterative generation process of $\mathbf{x}_0$. Instead, for every possible $\mathcal{H}$, we perform inference to obtain $\mathbf{x}_0$ and the corresponding $\varphi^*$ and directly obtain the optimal $\mathcal{H}^*$ since it is discrete. Here, we use the structural similarity index measure (SSIM) to approximate the $p(\mathcal{H} \mid \mathbf{x}_0, \mathbf{y}, \boldsymbol{\varphi})$. We measure the similarity of the measurement image and the estimated image operated by the forward function to obtain the optimal $\mathcal{H}^*$.

---

**Algorithm 1** Solving Inverse Problem with Unspecified Forward Operator by UFODM

---

1: **Input**: Measurement $\mathbf{y}$, initial values of parameter $\boldsymbol{\varphi}$, the set of potential function forms $\mathcal{S}_\mathcal{H}$
2: **Output**: Restored image $\mathbf{x}_0$, the forward operator $\mathcal{H}^*$ and its parameter $\boldsymbol{\varphi}^*$
3: Inference $\mathbf{x}_{T'} \sim q\left(\mathbf{x}_{T'} \mid \mathbf{y}\right)$                                                      ▷ perturb input
4: **for** $t \leftarrow T' \ldots 1$ **do**
5:      Inference $\mathbf{x}_{t-1} \sim \mathcal{N}\left(\mu_\theta\left(\mathbf{x}_t\right) + r_t^2 \Sigma_\theta\left(\mathbf{x}_t\right) \nabla_{\mathbf{x}_t} \log p_t\left(\mathbf{y} \mid \mathbf{x}_t, \boldsymbol{\varphi}, \mathcal{H}\right), \Sigma_\theta\left(\mathbf{x}_t\right)\right)$ .
6:                                        ▷ approximated by $p_\theta(\mathbf{x}_{t-1} \mid \mathbf{x}_t, \mathbf{y}, \boldsymbol{\varphi}, \mathcal{H})$
7:      **for** $i \leftarrow N \ldots 1$ **do**
8:          Inference $\boldsymbol{\varphi}^* = \arg\min_{\boldsymbol{\varphi}} \|\mathcal{H}_{\boldsymbol{\varphi}}(\hat{\mathbf{x}}_t) - \mathbf{y}\|_2 + R_{\boldsymbol{\varphi}}(\boldsymbol{\varphi})$
9:                                        ▷ the MAP estimation of $p_\theta(\boldsymbol{\varphi} \mid \mathbf{y}, \hat{\mathbf{x}}_t, \mathcal{H})$
10:          Inference $\mathbf{x}_{t-1} \sim \mathcal{N}\left(\mu_\theta\left(\mathbf{x}_t\right) + r_t^2 \Sigma_\theta\left(\mathbf{x}_t\right) \nabla_{\mathbf{x}_t} \log p_t\left(\mathbf{y} \mid \mathbf{x}_t, \boldsymbol{\varphi}^*, \mathcal{H}\right), \Sigma_\theta\left(\mathbf{x}_t\right)\right)$ .
11:                                        ▷ approximated by $p_\theta(\mathbf{x}_{t-1} \mid \mathbf{x}_t, \mathbf{y}, \boldsymbol{\varphi}^*, \mathcal{H})$
12:      **end for**
13: **end for**
14: Inference $\mathcal{H}^* = \arg\max_\mathcal{H} p(\mathcal{H} \mid \mathbf{x}_0, \mathbf{y}, \boldsymbol{\varphi}^*)$       ▷ the MAP estimation of $p(\mathcal{H} \mid \mathbf{x}_0, \mathbf{y}, \boldsymbol{\varphi}^*)$
15: **return** $\mathbf{x}_0, \mathcal{H}^*, \boldsymbol{\varphi}^*$

---

We refer to our approach as *UFODM*, which is formally presented in Algorithm 1. Compared to the Bayesian approach for the blind inverse problem (i.e., GibbsDDRM) (Murata et al., 2023), in addition to the different approximate inference algorithms, more importantly, UFODM makes a step towards solving the inverse problem with unspecified forward operator, which is challenging and realistic (see more details in Sec. 2).

# 4 EXPERIMENTS

## 4.1 SETUP

**Datasets** The Flickr Face High Quality (FFHQ) dataset (Karras et al., 2019) comprises 70,000 high-resolution human face images at $1024 \times 1024$ pixels. The Animal Faces-HQ (AFHQ) dataset (Choi et al., 2020) includes 15,000 high-quality animal face images at $512 \times 512$ pixels. The ImageNet validation dataset (Russakovsky et al., 2015) boasts a diverse array of 50,000 images spread across 1,000 distinct classes. In accordance with the experimental protocol of BlindDPS and GibbsDDRM, we select 1,000 validation images from FFHQ, 500 test images from the dog class in AFHQ, and 1,000 images from ImageNet validation set at a resolution of $256 \times 256$ pixels.

**Metrics** In accordance with the evaluation protocols of BlindDPS (Chung et al., 2023a) and GibbsD-DRM (Murata et al., 2023), we employ three quantitative measurement metrics: 1) Frechet Inception Distance (FID) (Heusel et al., 2017), 2) Learned Perceptual Image Patch Similarity (LPIPS) (Zhang et al., 2018), and 3) Peak Signal-to-Noise-Ratio (PSNR).

**Forward Operator** We carry out experiments involving three most representative forward operators: 1) motion blur, 2) JPEG compression, and 3) low resolution. Regarding motion blur, we employ the open-source codebase[1] to generate $64 \times 64$ blur kernels with an intensity value of 0.5. We set the JPEG compression parameter (quality factor) to 10, and the downsampling factor to 4 for low resolution In addition, we add Gaussian noise with $\sigma_y = 0.02$ for all inverse problems.

**Baselines** We classify related works into several categories for comparison with UFODM. The initial category encompasses supervised learning-based methods, such as MPRNet (Zamir et al., 2021) and DeblurGANv2 (Kupyn et al., 2019). Image-level prior-based methods such as Pan-DCP (Pan et al., 2017) and SelfDeblur (Ren et al., 2020) form the subsequent category. The final category involves diffusion models-based methods, including BlindDPS (Chung et al., 2023a) and GibbsDDRM (Murata et al., 2023).

**Experimental settings** We employ pre-trained checkpoints from DPS (Chung et al., 2023b), P2 weighting (Choi et al., 2022), and Guided Diffusion (Dhariwal & Nichol, 2021) for FFHQ, AFHQ, and ImageNet respectively, with no fine-tuning or adaptation. We adhere to the hyper-parameters of GibbsDDRM to ensure a fair comparison. Specifically, for our method, UFODM, we set $T' = 65$, the

---

[1] `https://github.com/LeviBorodenko/motionblur`

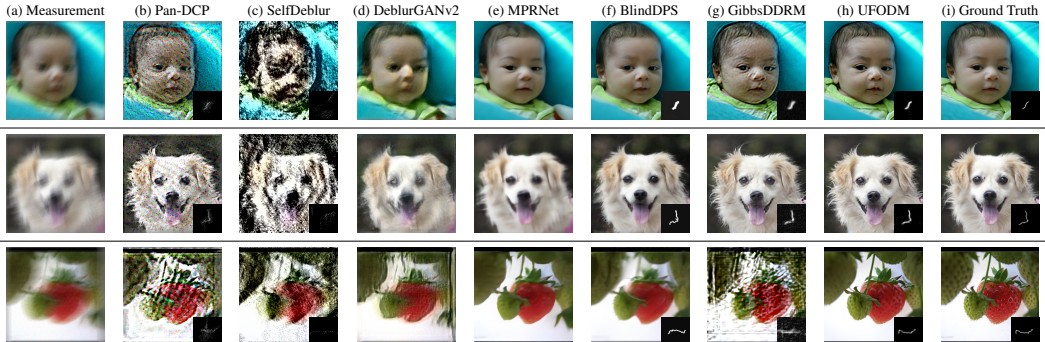

Figure 2: Visualization of blind motion deconvolution on FFHQ (first row), AFHQ (second row) and ImageNet (third row) datasets: (a) measurement, the sole input to the inverse problem solver, (b) Pan-DCP Pan et al. (2017), (c) SelfDeblur Ren et al. (2020), (d) DeblurGANv2 Kupyn et al. (2019), (e) MPRNet Zamir et al. (2021), (f) BlindDPS Chung et al. (2023a), (g) GibbsDDRM Murata et al. (2023), (h) UFODM (our method), (i) ground truth image and kernel for reference.

number of steps is determined at 65. The number of the iterations $N$ is set to 3 in each timestep. The number of iterations and the step size for the inference of $\varphi$ are fixed at 100 and 1e-4, respectively. The blur kernel is initialized with a Gaussian blur kernel. For the parameter $\varphi$, we employ a L1 regularization, and the coefficient weight is set to 1.

## 4.2 BENCHMARK EVALUATION: BLIND MOTION DECONVOLUTION

In this section, we select motion blur as the known forward operator for benchmarking. Thus, the only two remaining unknown variables are $\mathbf{x}_0$ and $\varphi$. The objective is to restore the original clean image $\mathbf{x}_0$ and estimate the forward function parameter $\varphi$ solely from the motion-blurred measurement, without the availability of the kernel parameter.

Table 1 presents the quantitative outcomes of blind deconvolution applied to the FFHQ, AFHQ, and ImageNet datasets. Our approach, enhanced with the time-travel trick (Wang et al., 2022) (denoted as ours*), optimizes image quality despite necessitating additional computational resources and latency. The proposed method, UFODM, outperforms all comparable studies in FID and LPIPS measurements, with the exception of the FID score from BlindDPS. Notably, the FID metric gauges the quality of restored images, while LPIPS evaluates the fidelity to the original ground truth images. BlindDPS records the highest FID score, with our method securing the second position. GibbsDDRM reported a similar observation, conjecturing that BlindDPS might generate more than required for noisy observations, potentially compromising its fidelity to the input image. The supervised model, MPRNet, achieves the highest PSNR, while our approach records the third highest PSNR on the FFHQ dataset and second highest PSNR on the AFHQ and ImageNet dataset. Additionally, UFODM yields superior FID and LPIPS scores compared to MPRNet, suggesting that the images generated by our method exhibit enhanced visual quality and fidelity.

Figure 2 illustrates the qualitative results of blind deconvolution applied to the FFHQ, AFHQ and ImageNet datasets. While MPRNet achieves the highest PSNR value, a closer examination reveals that the restored images lack fine details and exhibit a degree of blurriness compared to the ground truth. Additionally, parameter-estimated methods (e.g. Pan-DCP, SelfDeblur) yield blurry reconstructions due to inaccurate blur kernel estimation and are less robust in handling additional noise. For all the datasets, the three diffusion model-based approaches successfully recover the overall shape of motion blur parameter $\varphi$, yielding generated images with global visual content closely resembling the ground truth. However, subtleties matter. Our method, UFODM, provides the most accurate kernel estimation and photorealistic restoration. It is important to note that our approach outperforms all baselines in both FID and LPIPS metrics. This is further supported by our qualitative results, demonstrating that our method produces images with superior perceptual similarity and quality when compared to the ground truth.

Table 1: Quantitative assessment of blind motion deblurring on FFHQ, AFHQ, and ImageNet at a resolution of $256 \times 256$ pixels. The best performing method is marked in **bold**, while the runner-up is underlined.

| Method | FFHQ ($256 \times 256$) | | | AFHQ ($256 \times 256$) | | | ImageNet ($256 \times 256$) | | |
|---|---|---|---|---|---|---|---|---|---|
| | FID ↓ | LPIPS ↓ | PSNR ↑ | FID ↓ | LPIPS ↓ | PSNR ↑ | FID ↓ | LPIPS ↓ | PSNR ↑ |
| MPRNet (Zamir et al., 2021) | 62.92 | 0.211 | **27.23** | 50.43 | 0.278 | **27.02** | 87.56 | 0.332 | **24.70** |
| DeblurGANv2 (Kupyn et al., 2019) | 141.55 | 0.320 | 19.86 | 156.92 | 0.429 | 17.64 | 166.77 | 0.444 | 18.18 |
| Pan-DCP (Pan et al., 2017) | 239.69 | 0.653 | 14.20 | 185.40 | 0.632 | 14.48 | 205.87 | 0.651 | 12.45 |
| SelfDeblur (Ren et al., 2020) | 283.69 | 0.859 | 10.44 | 250.20 | 0.840 | 10.34 | 230.85 | 0.763 | 9.89 |
| BlindDPS (Chung et al., 2023a) | **29.49** | 0.281 | 22.24 | **23.89** | 0.338 | 20.92 | **63.96** | 0.361 | 20.65 |
| GibbsDDRM (Murata et al., 2023) | 38.71 | 0.115 | 25.80 | 48.00 | 0.197 | 22.01 | 144.28 | 0.470 | 19.29 |
| UFODM (ours) | 34.49 | 0.126 | 24.15 | 27.12 | 0.166 | 22.68 | 78.21 | 0.336 | 19.43 |
| UFODM (ours*) | 30.06 | **0.107** | 25.20 | 25.81 | **0.159** | 22.95 | 64.93 | **0.295** | 20.77 |

## 4.3 CHALLENGE EXPLORATION: INVERSE PROBLEM WITH UNSPECIFIED FUNCTION

In this section, we select 1) motion blur, 2) JPEG compression, and 3) low resolution as the three most representative forward functions for benchmarking. As the function form should not be predefined for practical applications, we thus randomly apply one of three forward operators to the given input images from FFHQ, AFHQ, and ImageNet, setting up the challenge of solving inverse problems with unspecified forward operator. It is important to note that JPEG compression is among the most common nonlinear and non-differentiable forward operators. In this situation, BlindDPS requires the function to be differentiable, while GibbsDDRM requires the function to be linear to utilize singular value decomposition (SVD). Furthermore, BlindDPS heavily relies on a data-driven approach to establish the diffusion prior, resulting in an untenable computational cost to support each forward operator. As a result, our method, UFODM, stands as the sole generic diffusion model-based inverse problem solver capable of accommodating all three forward operators.

Table 2 benchmarks the complex inverse problems with unspecified forward operator, where the input image is corrupted by an unspecified forward operator. To the best of our knowledge, UFODM is the pioneering approach designed to tackle this practical challenge. We showcase our method by using various techniques to deduce the forward operator $\mathcal{H}$. The high values of FID in Table 2 is due to the dataset size, as we utilize 1/3 of each dataset for different forward operators. In this study, we employ the Structural Similarity Index (SSIM) to calculate the match between the given corrupted input and the inferred image, using the generated clean image and the estimated forward operator. It's worth noting that the accuracy of function form selection is 96.4%, 85.8% and 61.3% on the FFHQ, AFHQ, and ImageNet datasets, respectively. These results highlight the effectiveness of UFODM's MAP inference for $\mathcal{H}$.

We also extend our analysis of the proposed framework to a more intricate setting involving a larger problem set comprising six distinct forward operators. This comprehensive set encompasses blind deconvolution, JPEG restoration, super resolution, inpainting, as well as two challenging scenarios involving combinations of forward operators: inpainting paired with super resolution, and deblurring coupled with super resolution. To validate the efficacy of UFODM, we present qualitative results obtained on the ImageNet, AFHQ, and FFHQ datasets, as illustrated in Figure 3. These results vividly showcase UFODM's promising potential in handling inverse problem with unspecified forward operator, and highlight areas for future improvement in the model.

## 5 DISCUSSION

In this study, we introduce an innovative framework UFODM that addresses the considerable challenge associated with the unspecified forward operator in diffusion models for inverse problems. By strategically selecting the appropriate forward measurement, our methodology facilitates simultaneous estimation of both the restored image and the forward operator, significantly enhancing the practical applicability of diffusion models in real-world situations. Our framework has demonstrated

Table 2: Quantitative evaluation of UFODM (our approach) for addressing inverse problems in uncontrolled settings on FFHQ, AFHQ and ImageNet at a resolution of $256 \times 256$ pixels.

| Dataset | Blind Deconvolution | | | JPEG Restoration | | | Super Resolution | | |
|---|---|---|---|---|---|---|---|---|---|
| | FID ↓ | LPIPS ↓ | PSNR ↑ | FID ↓ | LPIPS ↓ | PSNR ↑ | FID ↓ | LPIPS ↓ | PSNR ↑ |
| FFHQ | 55.78 | 0.119 | 24.45 | 56.43 | 0.104 | 28.12 | 60.04 | 0.103 | 29.04 |
| AFHQ | 33.38 | 0.142 | 23.52 | 31.78 | 0.153 | 25.29 | 28.29 | 0.129 | 27.55 |
| ImageNet | 129.73 | 0.342 | 19.24 | 156.94 | 0.345 | 21.91 | 110.53 | 0.320 | 21.21 |

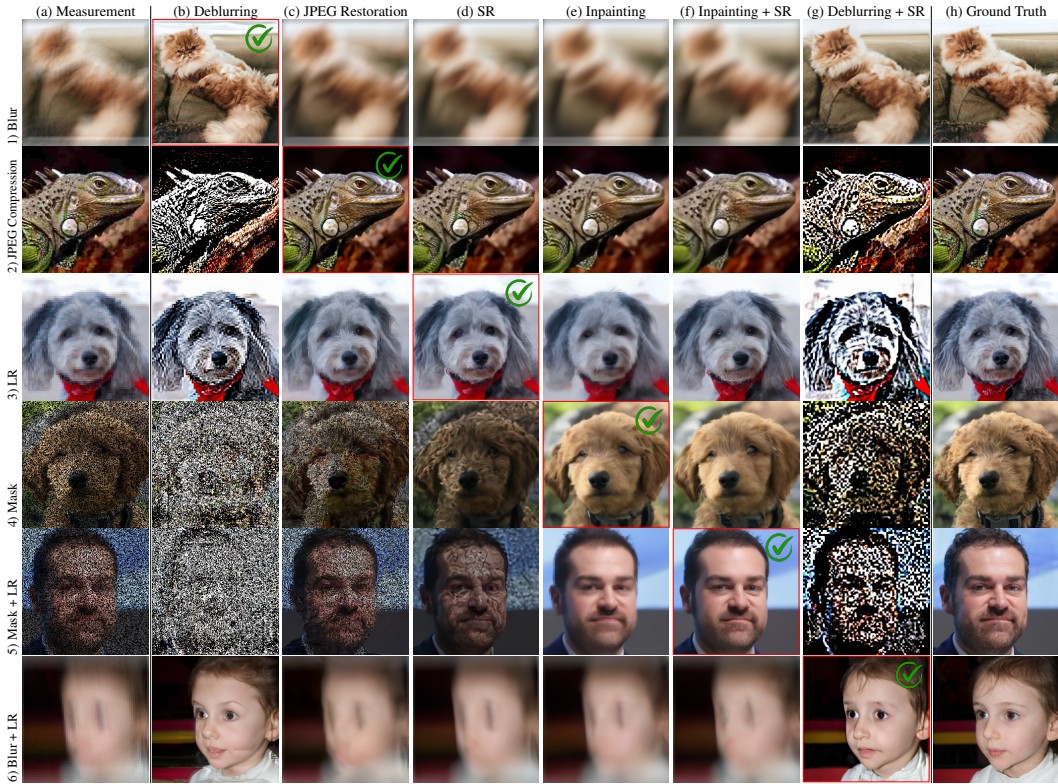

Figure 3: Illustration of the inverse problem with unspecified forward operator using the ImageNet, AFHQ, and FFHQ dataset. Given (a) measurements of 1) motion blur, 2) JPEG compression, 3) LR(low resolution), 4) mask, 5) mask + LR, and 6) blur + LR, UFODM autonomously selects the suitable inverse problem solver from (b) deblurring, (c) JPEG restoration, (d) SR (super resolution), (e) Inpainting, (f) Inpaint + SR, and (g) deblurring + SR, thereby expanding the applicability of diffusion models in the wild. Our approach produces images (indicated by a green check mark at the upper right corner) exhibiting superior perceptual similarity and quality compared to (h) ground truth.

remarkable versatility and efficacy in solving a range of generic inverse problems. This holds true even when the form of the forward function remains undefined.

**Future Direction** This research lays the groundwork for numerous future studies. The effectiveness and resilience of our methodology depend on the chosen forward measurement selection strategy, meriting further exploration of alternative approaches in future research. While our framework shows promising results in six benchmark tasks, its applicability to a wider range of inverse problems is still undetermined, suggesting a valuable avenue for future work. By advancing the use of diffusion models in inverse problems, our study constitutes a major advance in image restoration.

**Ethics statement**   While our work aims to exploit generative models to address inverse problems, we must remain cognizant of dataset bias. Since our diffusion model is entirely pre-trained on predefined data, biases present in the data could be mirrored or amplified by the trained model. Having been trained on biased data, the diffusion model is prone to projecting the given measurement towards these biases, potentially losing important or sensitive attributes of the input image in the process.

**Reproducibility statement**   We have made the following efforts to facilitate reproducibility of our work. (1) Our experiments are conducted on publicly available datasets and model checkpoints (see 4.1). (2) We include a detailed description of our algorithm in Algorithm 1. (3) We include our source code in supplemental materials.

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

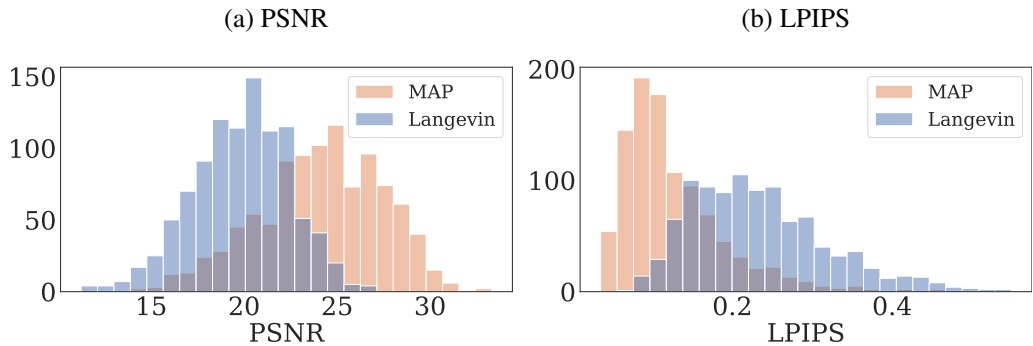

Figure 4: Histograms of blind image deblurring results on FFHQ (256 × 256) dataset obtained from different estimation strategies for $\varphi$. MAP: Maximum a posteriori (MAP) estimation; Langevin: the estimation method from GibbsDDRM Murata et al. (2023).

## A    ADDITIONAL ANALYSIS

**Comparison between MAP and Langevin**    We performed an analysis of the two-sample strategy, namely Maximum a Posteriori (MAP) estimation and Langevin, the latter being an estimation method derived from GibbsDDRM Murata et al. (2023). Figure 4 presents histograms of blind image deblurring results from the FFHQ dataset utilizing different $\varphi$ estimation strategies. The results indicate that the MAP strategy surpasses the Langevin method in both PSNR and LPIPS metrics, thereby yielding a more accurate estimation of the parameter $\varphi$.

**Efficiency Analysis**    We thoroughly evaluate the computational efficiency of our proposed approach. To provide a precise assessment of the incurred computational overhead, we conducted the blind motion deconvolution experiments using an NVIDIA RTX 2080ti GPU, measuring the wall-clock time required for the reconstruction of a single image.

Furthermore, in comparison to diffusion-based methods like BindDPS (180.22 seconds) and GibbsDDRM (262.68 seconds), our UFODM exhibits remarkable efficiency, completing the image reconstruction task in just 94.48 seconds. Our approach not only excels in terms of speed but also delivers superior performance. This underscores the efficiency and effectiveness of our method, indicating its potential for real-world applications. When comparing our results against well-established algorithms in the field, parameter-estimated methods like Pan-DCP (175.96 seconds) and SelfDeblur (178.99 seconds) demonstrate substantially longer processing times. DeblurGANv2 (2.14 seconds) and MPRNet (1.91 seconds) also show faster processing times, but they may lag behind in terms of image quality and overall performance.

## B    ADDITIONAL RESULTS

**Additional Visualizations**    Additional experimental outcomes regarding blind deconvolution are presented in Figure 5. This figure provides a visual representation of blind deconvolution applied to the FFHQ, AFHQ and ImageNet datasets. Our proposed method, UFODM, excels in delivering the most precise kernel estimation and photorealistic restoration. This claim is backed by our qualitative results, as our method produces images demonstrating heightened perceptual similarity and superior quality when compared to the ground truth.

Figure 6 portrays the progression of $\mathbf{x}_t$ and $\varphi$. Notably, the estimated $\varphi$ closely approximates the ground truth kernel at a very early stage, courtesy of our initialization strategy, even while $\mathbf{x}_t$ remains significantly noisy.

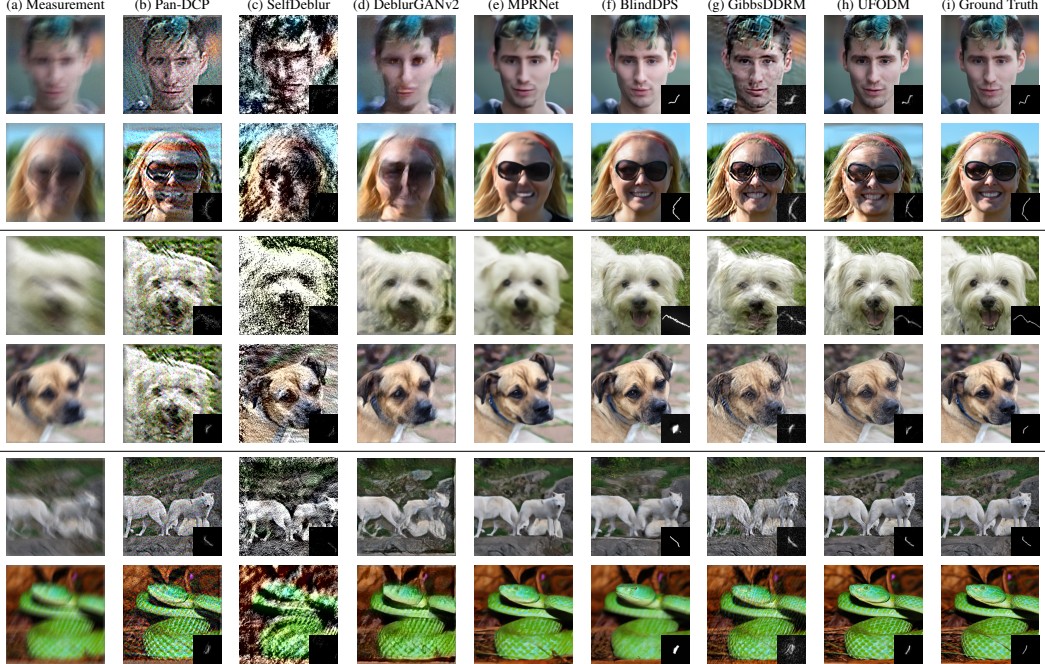

Figure 5: Additional visualizations of blind deconvolution on FFHQ, AFHQ, and ImageNet datasets: (a) measurement, the sole input to the inverse problem solver, (b) Pan-DCP Pan et al. (2017), (c) SelfDeblur Ren et al. (2020), (d) DeblurGANv2 Kupyn et al. (2019), (e) MPRNet Zamir et al. (2021), (f) BlindDPS Chung et al. (2023a), (g) GibbsDDRM Murata et al. (2023), (h) UFODM (our method), (i) ground truth image and kernel for reference.

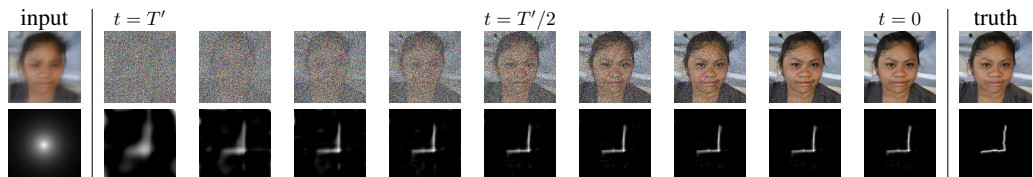

Figure 6: Visualization of UFODM with simultaneous estimation of both restored image and forward operator for blind motion deconvolution on FFHQ dataset.

