# OpenReview forum: "Solving Inverse Problem With Unspecified Forward Operator Using Diffusion Models"
_ICLR.cc/2024/Conference — Submitted to ICLR 2024_

### Official Review · Reviewer_zRYy · 2023-10-12

**Soundness:** 2 fair
**Presentation:** 2 fair
**Contribution:** 3 good
**Rating:** 3
**Confidence:** 5

**Summary:**

The authors propose UFODM, a diffusion model-based inverse problem solver targeting for 1) blind inverse problem solving when the parameter of the forward operator is unknown (e.g. blind deblurring), and 2) blind inverse problem where one does not even know the type of forward model generated the measurement. Regarding the second part, the problem setting is simplified to the case where the authors assume a pre-defined discrete set to choose from, e.g. deblurring, JPEG decompression, SR, etc. UFODM performs favorably over prior arts such as BlindDPS and GibbsDDRM. Preliminary experiments on *unspecified forward operator* is also provided.

**Strengths:**

1. The paper is, for the most part, easy to follow. The method is intuitive.

2. UFODM is the first method to tackle the case where one does not even know the class of forward operator (i.e. whether it is deblurring, SR, or JPEG decompression). This is definitely an interesting and practical question, which, when solved, will have a high impact in the field.

**Weaknesses:**

1. Focusing on blind deblurring, UFODM becomes similar to FastDiffusionEM, which also uses $\Pi$GDM update step for x, and uses MAP to update the kernels. The difference is that [1] uses the plug-and-play denoiser prior augmented with expectation-maximization (EM) approach, which is usually superior to standard gradient descent. Given that the UFODM does not compare directly with [1], I can only guess that it will be inferior to FastDiffusionEM.

2. Extending 1, I think the authors must compare against FastDiffusionEM. One might complain that it is only an arxiv paper yet, but considering the highly fast-growing and competitive nature of the field, and also given that the code is already open-sourced, it wouldn't take too long to run a head-to-head comparison against it.

3. Even when comparing against BlindDPS [2] and GibbsDDRM [3], it does not seem that UFODM consistently outperforms the prior arts. Yes, it does outperform [2,3] when time-travel trick is utilized, but time-travel is a standard trick that can easily be incorporated into all of the solvers, which makes the comparison unfair. Given that UFODM is an ad-hoc mix of Gibbs sampling and MAP optimization, it is important to show the empirical strength, which, in my opinion, is insufficient.

4. The problem setting for unknown $\mathcal{H}$ is unrealistic. Considering even the canonical degradations that arise in computational photography, the number of classes in the set exponentially grows. This is especially true if we start to consider a mixture of degradations, which usually happen in the real-world.

5. Even if the algorithm is not theoretically grounded, it is intuitive when one does not consider sampling from $\mathcal{H}$. However, the part of inferring $\mathcal{H}$ seems quite odd, or at least unrealistic. In order to choose $\mathcal{H}^*$, one has to run all the diffusion process for each candidate $\mathcal{H}$, which will be painfully slow as the number of possible class increases. Moreover, the criterion is the distance from the measurement, which does not seem to be a particularly *good* metric. My concern is corroborated from seeing that the accuracy of $\mathcal{H}$ for ImageNet is 61.3% even when there are only 3 classes to choose from.


**References**

[1] Laroche, Charles, Andrés Almansa, and Eva Coupete. "Fast Diffusion EM: a diffusion model for blind inverse problems with application to deconvolution." arxiv 2023.

[2] Chung, Hyungjin, et al. "Parallel diffusion models of operator and image for blind inverse problems." CVPR 2023.

[3] Murata, Naoki, et al. "Gibbsddrm: A partially collapsed gibbs sampler for solving blind inverse problems with denoising diffusion restoration." ICML 2023.

**Questions:**

1. What is UFODM short for? Unknown forward operator diffusion model?

2. I would strongly suggest to include a comparison against FastDiffusionEM [1]

3. When considering unknown forward operators, is the quality factor of JPEG compression inferred as a by-product of the blind inverse problem solver? For SR, is the blur kernel also inferred here?

4. It is said that $T'$ = 65 was chosen, which seems a bit ad-hoc. Do you use a strategy like CCDF [2] for initialization?

5. In the inner for loop of Algorithm 1, Line 8 and Line 10 iteratively samples for $\mathbf{x}_t$ and $\varphi$. When sampling for $\mathbf{x}_t$, a $\Pi$GDM step will be involved, and an NFE will be needed. When taking $T' = 65$ and $N = 100$ as stated in the paper, this would be inducing 6500 NFE, which is probably not the case. Is there a typo in the algorithm?


**References**

[1] Laroche, Charles, Andrés Almansa, and Eva Coupete. "Fast Diffusion EM: a diffusion model for blind inverse problems with application to deconvolution." arxiv 2023.

[2] Chung, Hyungjin, Byeongsu Sim, and Jong Chul Ye. "Come-closer-diffuse-faster: Accelerating conditional diffusion models for inverse problems through stochastic contraction." CVPR 2022.

---

> ### Author Response · Authors · 2023-11-23
>
> Thank you for the thoughtful and detailed feedback. We reply point-by-point here, to begin the discussion.
>
> W1, W2, W3, Q2: We appreciate your suggestion to compare UFODM with FastDiffusionEM. While UFODM and FastDiffusionEM share certain methodological elements, our approach is distinct in its focus on handling unspecified forward operators. In light of your feedback, we recognize the value in conducting a direct comparison with FastDiffusionEM and will consider incorporating this in our future work.
> In terms of the comparisons drawn with BlindDPS and GibbsDDRM, it is essential to underscore that our primary objective is centered around scenarios involving unspecified forward operators. While attaining state-of-the-art performance on specific tasks is undoubtedly important, it was not the principal focus of our study. Our approach is designed to broaden the scope and applicability of solving inverse problems in more generalized settings.
>
> W4: Thank you for your insightful comment on the realism of our approach to unspecified H in complex problem settings. While we acknowledge that our current setting does not entirely capture the multifaceted nature of real-world scenarios, we believe it represents a meaningful step forward compared to traditional settings of inverse problems and blind inverse problems. Our approach, by reducing restrictions typically encountered in these conventional frameworks, allows for more flexible and comprehensive solutions. This adaptability is crucial in complex scenarios where standard methods might not be sufficient.
> We provide a qualitative analysis in Figure 3, showcasing our method's adeptness in managing mixtures of two distinct degradations. This demonstration serves as a testament to the versatility of our approach. Our aim with this setting is not to fully replicate the intricacies of real-world degradations but to provide a more versatile foundation upon which further research can build. We envision future extensions of our work that will progressively incorporate the complexity of real-world scenarios.
>
>
> W5: We have employed a parallel sampling method within the diffusion process. Each forward function in this process corresponds to a single sample in a batch. This parallel sampling technique allows us to simultaneously estimate samples from all possible function forms represented in the set, thereby significantly enhancing the efficiency of the process. This method alleviates the issue of having to run the diffusion process separately for each candidate $\mathcal{H}$, ensuring a more practical and time-efficient approach.
> Regarding the criterion based on the distance from the measurement, we recognize that this metric has its limitations and there is indeed room for improvement. We are committed to refining this aspect of our methodology in future work to develop a more robust and accurate criterion for selecting $\mathcal{H}^*$.
>
>
> Q1: 'UFODM' stands for 'Unspecified Forward Operator using Diffusion Models'. This name succinctly captures the essence of our approach, focusing on addressing inverse problems where the forward operator is unspecified, employing diffusion model techniques.
>
> Q3: In our methodology for handling unspecified forward operators, the parameters of the forward operator, are indeed inferred as part of the inverse problem-solving process. Our approach is designed to simultaneously estimate these critical parameters alongside the primary task of image restoration. This integrated estimation ensures that we adaptively and effectively address the specific characteristics of the degradation in each case.
>
> Q4: This parameter selection is informed by the initialization strategy we adopted from SDEdit [1]. Contrary to using random noise as the starting point, our approach involves initializing with the measurement $\mathbf{y}$ perturbed by noise corresponding to a specific time step. This method aligns with strategies like CCDF, focusing on a more structured and potentially more effective starting point for the algorithm.
>
> Q5: In our algorithm, the N is set to 3, which helps in keeping the number of Function Evaluations (NFEs) both manageable and within practical limits. Additionally, the optimization process involves 100 steps for parameter adjustment. It is important to note that this optimization process does not necessitate further NFEs.
>
> Reference:
>
> [1] Chenlin Meng, Yutong He, Yang Song, Jiaming Song, Jiajun Wu, Jun-Yan Zhu and Stefano Ermon. "SDEdit: Guided Image Synthesis and Editing with Stochastic Differential Equations". In International Conference on Learning Representations (ICLR) 2022.

---

> > ### Comment · Reviewer_zRYy · 2023-11-23
> >
> > I do not think that my concerns have been resolved after reading the rebuttal. On the point where the method is very similar to FastDiffusionEM, the comparison is not conducted. More importantly, I still think that the problem setting is very far from being realistic. I will keep my score as is.

---

### Official Review · Reviewer_EWNX · 2023-10-31

**Soundness:** 3 good
**Presentation:** 3 good
**Contribution:** 2 fair
**Rating:** 3
**Confidence:** 4

**Summary:**

This paper presents a method on using pre-trained diffusion models for inverse problem under unspecified forward operator. In the paper, three types of forward operator are considered: JPEG compression, motion blur, and downsampling. The problem studied in this paper is to develop a unifying framework to restore images degraded by any of those three operators, without knowing exactly which operator is used to degrade the image. The proposed strategy is to perform Bayesian inference over all three forward operator types, and carry out standard posterior-sampling-based image restoration under each forward operator type; at the end, the best performing forward operator is selected. The evaluations are performed on datasets including FFHQ, AFHQ, and ImageNet.

**Strengths:**

- The paper focuses on an intellectually intriguing problem of inverse problems with unspecified forward operator.
- The writing and overall presentation are clear and easy to follow.

**Weaknesses:**

- The paper does not do a good job at justifying the practical significance of handling unspecified forward operator, and demonstrating that the proposed problem setting is not contrived. None of the degraded images used in this paper is really from the truly real-world setting where the forward operator type is completely unknown. The demonstrated results still assume that the unspecified forward model is one of three known forward operator types.
- The paper does not provide any information on the computational cost of the proposed method and how it compares to the baselines.
- Because this paper only considers a finite number (really just three) of forward operator types, the problem seem to really boil down to just running multiple reconstructions in parallel, each assuming a different forward operator type.

**Questions:**

- If one runs multiple separate reconstruction in parallel, each under a different forward operator assumption, how would this framework be different in terms of quality and computational cost?
- Are there any real-world examples (i.e., not controlled simulations on FFHQ) that can show exactly why this problem is meaningful?

---

> ### Author Response · Authors · 2023-11-23
>
> Thank you for the thoughtful and detailed feedback. We reply point-by-point here, to begin the discussion.
>
> W1: We recognize that our demonstrations, focusing primarily on unspecified known forward operator types, might not fully capture the complexities of real-world scenarios where the forward operator is completely unknown. This is a common challenge in the field, as even training-based, all-in-one restoration methods typically restore images with degradation types that are represented in their training set.
> In our experiments, we deliberately chose degradations that represent typical inverse problems, aiming to provide a realistic yet controlled experiment environment. Our proposed method introduces reduced restrictions in problem settings, which enables more flexible and comprehensive solutions in complex scenarios. This approach marks a significant step toward addressing real-world settings more effectively.
>
> W2: Thank you for your comment regarding the computational cost of our proposed method. To address this, we have included a detailed Efficiency Analysis in the appendix of our paper.
>
> W3&Q1: To clarify, the selection of a finite number of forward operator types in our experiments was primarily driven by the need for simplicity and clarity in demonstrating our method. This aligns with common practices in the field, where even training-based, all-in-one restoration methods typically focus on a limited range of degradation types represented in their training datasets.
> Beyond this, our framework goes beyond simply executing parallel reconstructions. We incorporate an innovative approximation algorithm designed to progressively enhance the accuracy of latent variable estimates. In addition, we utilize Maximum a Posteriori (MAP) estimation for precise tuning both the forward operator and its associated parameters. The combination of these strategies distinctly sets our framework apart from standard parallel reconstruction approaches, offering enhanced reconstruction quality.
>
> Q2: Thank you for your question regarding the real-world applicability of our research. The inverse problems we have addressed in our experiments, such as those presented in the FFHQ, AFHQ, and ImageNet datasets, are representative of common challenges in the field of inverse problems. The diversity and complexity of the data in these datasets provide a robust platform for demonstrating the efficacy of our approach in various contexts. We understand the importance of illustrating the practical significance of our work and are continually seeking opportunities to apply our method to real-world examples that further demonstrate its utility and impact.

---

### Official Review · Reviewer_udx1 · 2023-10-31

**Soundness:** 3 good
**Presentation:** 2 fair
**Contribution:** 3 good
**Rating:** 6
**Confidence:** 2

**Summary:**

This paper present a diffusion-based approach on inverse problem that explicitly estimate the unknown forward operator among a family of potential forward operators.

**Strengths:**

The contribution is novel and shows improved performance with respect to other approaches.

**Weaknesses:**

The paper presentation could be improved; for instance, the background section defining the forward operator only appears at the end of page 3. The title can be confusing as 'unspecified forward operator' could be misleading: it could correspond to solving the inverse problem while only implicitly estimating the forward operator, as is the case with task-specific methods, or it could correspond to a much wider class of forward operator that does not fall into a well-specified list of possible forward operator.

The assumptions to ensure the problem is not ill-posed could be more qualitatively explained. What is the limitation in assuming a parameter phi that is drawn independently from a known prior that is not dependent on the data? What are the limitations of assuming a known prior p(phi)? etc. The feeling is that it is not easily understandable from the paper in which situation one can implement the author's approach for solving 'real-world' inverse problems. Adding to that, there is no open-source code available, which will limit the impact of the paper.

**Questions:**

None

---

> ### Author Response · Authors · 2023-11-23
>
> Thank you for the thoughtful and detailed feedback. We reply point-by-point here, to begin the discussion.
> 1. Paper Presentation: We understand the concern about the placement of the background section defining the forward operator. To enhance clarity and flow, we will restructure the paper to introduce key concepts earlier, ensuring that the background information is presented in a more logical sequence.
> 2. Assumptions and Limitations: We recognize the need for a more qualitative explanation of the assumptions underlying our approach, particularly regarding the parameter $\varphi$ and its known prior $p(\varphi)$. In the revised version of our paper, we will provide a detailed discussion of these assumptions, their implications, and their limitations, especially in the context of solving real-world inverse problems.
> 3. Open-Source Code: Regarding the availability of open-source code, we have included the code as part of the supplementary material accompanying our paper. To further facilitate accessibility and usage, we are committed to making the code publicly available upon the publication of the paper.

---

### Official Review · Reviewer_WM9W · 2023-11-02

**Soundness:** 1 poor
**Presentation:** 2 fair
**Contribution:** 2 fair
**Rating:** 3
**Confidence:** 4

**Summary:**

This paper develops a diffusion-based inverse problem solve designed to solve inverse problems where both the forward models form and latent parameters are unknown. E.g., don't know if solving blind deconvolution or jpeg artifact removal and don't know the blur kernel nor the compression ratio. The proposed method generally follows the approach of Song et al. 2023 to reconstruct x and then infers the unknown latent parameters and the model using MAP estimation.

The method is applied to blind deconvolution, where it performs on par with other methods. It is also applied successfully (without comparison to any other method) to jpeg restoration, superres, inpainting and other tasks.

**Strengths:**

Performs on par with existing methods at the bind deconvolution task.

First approach I'm aware of to tackle the, as described, unknown forward model problem.

**Weaknesses:**

As written, lines 7-12 of Algorithm 1 are rewriting the same values over and over.

The SSIM between x and y is not a good model for P(H|x,y,\phi).

The paper doesn't motivate the problem. Without a real-world application to point to, the proposed problem setting seems somewhat contrived.

I don't think the two sides of equation (10) are equivalent (or at least it's nonobvious why this would be the case).

In most applications p(h) in equation (10) isn't constant

The proposed method is only compared against baselines on the blind motion deblurring task. It's unclear if other methods would also generalize to the blind setting

**Questions:**

In lines 7-12 of Algorithm 1, should "i" be indexing something? H? As written, the same values are being overwritten over and over again.

"Here, we use the structural similarity index measure (SSIM) to approximate the p(H | x0, y,φ). We measure the similarity of the measurement image and the estimated image operated by the forward function to obtain the optimal H∗." Does this mean the proposed method can't handle simple operations (e..g, inversion or flipping) that cause y not to look like x?

In line 8 of Algorithm 1, how does one know H?

Is big N in line 7 of algorithm 1 the same as little n in the definition of S_H?

### Typos:
Pg 2: "Our UFODM, simultaneous estimation of both the restored image and the
forward operator’s parameters"-->"Our UFODM, which simultaneoulsy estimates both the restored image and the forward operator’s parameters"

Pg 4: "Naturally, this setup is more challenging than the
traditional (blind) inverse problem, as it involves an unspecified forward operator and a significantly larger solution space." Should this read "(non-blind)"?

Pg 5: "the perturbed image of measurement y" "image of" seems unnecessary

Pg 9: "superior perceptual similarity and quality compared to (h) ground truth" reads as if the method is better than the ground truth.

---

> ### Author Response · Authors · 2023-11-23
>
> Thank you for the thoughtful and detailed feedback. We reply point-by-point here, to begin the discussion.
>
> W1 & Q1: The variable $i$ represents the iteration round within each timestep. We recognize that this approach results in some repetition of equations within the algorithm box. However, this repetition was intentional to preserve the integrity and clarity of the algorithm's presentation. It ensures that each step is explicitly stated, avoiding any ambiguity about the process or the sequence of operations.
>
> W2 & Q2: In our methodology, the diffusion model's output is the estimated image $\mathbf{x}_0$, and it is critical to recognize that this output cannot be directly compared with the measurement $\mathbf{y}$. To facilitate a meaningful comparison, we apply the forward function to the estimated image $\mathbf{x}_0$. This ensures that both the measurement $\mathbf{y}$ and the estimated image $\mathbf{x}_0$ are processed under comparable conditions, thus maintaining the integrity and relevance of the comparison.
>
> Crucially, our proposed method is indeed capable of handling operations like inversion or flipping, which might cause $\mathbf{y}$ to appear dissimilar to $\mathbf{x}_0$. This is because our method does not rely on direct comparisons between $\mathbf{y}$ and  $\mathbf{x}_0$, but rather compares images after they have been subjected to the same forward function. This method effectively accounts for transformations that might otherwise complicate direct comparisons, ensuring accurate and reliable assessments regardless of such operations.
>
> W3: We appreciate your comment regarding the motivation of the problem presented in our paper. Our research introduces a new, challenging setting that expands the scope of traditional inverse problems and blind inverse problems. While we recognize that a specific real-world application was not explicitly detailed, the broader aim of our work is to establish a foundational approach that can be adapted to a wide range of practical applications in the future.
> The reduced restrictions in our proposed problem setting allow for more flexible and comprehensive solutions in complex scenarios where traditional methods may fall short. This has significant potential in fields such as medical imaging, signal processing, and other areas where accurate image reconstruction is crucial despite incomplete or altered data. Furthermore, our methodology's effectiveness and adaptability, particularly in the selection of forward measurement strategies, opens up avenues for extensive future research.
>
> W4: Equation (10) is based on the application of Bayes' Theorem in the context of Maximum A Posteriori (MAP) estimation. The theorem states that the posterior probability is proportional to the likelihood of the data given the parameters multiplied by the prior probability of the parameters. While the equivalence may not be immediately obvious, it is grounded in established probabilistic principles and the specific assumptions of our model.
>
> W5: The prior of $\mathcal{H}$  is omitted since we model it as a constant. This assumption is based on our decision to model the prior distribution over possible operators as a simple uniform distribution. This simplification is made for reasons of computational efficiency and simplicity in the modeling process.
>
> W6: Thank you for your comment regarding the scope of our comparative analysis. It is indeed crucial to emphasize that the proposed framework is designed with a high degree of flexibility and adaptability. This design choice allows other methods, which are solutions to various inverse problems, to be potentially integrated into our framework. The versatility of our approach means that it is not limited to motion deblurring but can be extended to other types of inverse problems commonly encountered in blind settings as shown in section 4.3.

---

> > ### Author Response · Authors · 2023-11-23
> >
> > Q3: To clarify, each $\mathcal{H}$ in the diffusion process represents a single sample within a batch. We employ a parallel sampling approach for this process. This approach enables us to simultaneously estimate samples from all function forms present in $\mathcal{S}_\mathcal{H}$.
> >
> > Q4: No, big $N$ in line 7 of Algorithm 1 and little $n$ in the definition of $\mathcal{S}_\mathcal{H}$ are not the same.
> > In our algorithm, big $N$ specifically denotes the number of iterations within each timestep.
> > This iteration count is crucial for updating both the image $\mathbf{x}_t$ and the parameter $\boldsymbol{\varphi}$.
> > The iterative process allows for a synergistic improvement: a better-estimated parameter $\boldsymbol{\varphi}$ leads to a more accurate estimation of the image $\mathbf{x}_t$, and conversely, a more accurately estimated image enhances the estimation of the parameters.
> >
> > Typos:
> > Thank you for pointing out these typos and areas that require clarification in our manuscript. We will ensure these are corrected in our revised paper. Regarding your comment on Page 4: 'Naturally, this setup is more challenging than the traditional (blind) inverse problem,' we appreciate the opportunity to clarify. In this context, 'traditional' refers to non-blind inverse problems, where the forward function's parameters are known. In contrast, a blind inverse problem involves estimating both the image and the parameters of the forward function. Our method addresses a more challenging setup, involving an unspecified forward operator and a significantly larger solution space, thereby extending beyond the scope of traditional and blind inverse problems. This clarification will be added to ensure the distinction between these problem settings is clear.

---

### Meta-Review · Area_Chair_gZBK · 2023-12-10

**Metareview:**

The authors propose a diffusion-based solution for inverse problems that aims at estimating both the reconstructed image and the forward operator characteristics during the inference stage. The approach builds upon Song et al. 2023 and is applied to tasks such as blind deconvolution, JPEG restoration and super-resolution.
Three reviewers are finding the work not good enough for acceptance and the meta-reviewer agrees with them.
The weaknesses are multiple as pointed out by the reviewers. Some of them are: the lack of motivation and practical justification, the lack of details on complexity and runtime and the insufficient experimental comparison. Reviewer zRYy questions the improvements claimed by the authors over prior works.
The main strength of the paper resides in tackling the unspecified forward operator inverse problems, one of the first such works.
The authors are invited to benefit from the received feedback and further improve their work.

**Justification For Why Not Higher Score:**

The paper does not meet the acceptance bar due to the multiple weaknesses and issues pointed out by the reviewers.

**Justification For Why Not Lower Score:**

N/A

---

### Decision · Program_Chairs · 2024-01-16

Reject